# A Feedback System Supporting Students Approaching a High-Level Programming Course

**Jong-Yih Kuo** [1,*] **, Hui-Chi Lin** [1] **, Ping-Feng Wang** [2] **and Zhen-Gang Nie** [3]

1   Department of Computer Science and Information Engineering, National Taipei University of Technology, Taipei 106344, Taiwan; huchlin@mail.ntut.edu.tw
2   Institute for Information Industry, Taipei 10622, Taiwan; pfwang@iii.org.tw
3   School of Information and Electronics Beijing, Institute of Technology, Beijing 100081, China; zhengang.nie@bit.edu.cn
*   Correspondence: jykuo@ntut.edu.tw

**Abstract:** This study analyzes the mistakes students are prone to make in programming and uses the GDB and Valgrind tools to implement dynamic analysis techniques for their eventual application to programs created by students. In the analysis process, spectral error localization technology is used to strengthen the dynamic analysis to find errors more accurately. The analyzed results are sorted and corresponding feedback is given to students in order for them to better understand the content of errors when revising the program and classifying and counting the types of errors made. This study sorts mistakes frequently made by students and topics in which students are likely to make certain mistakes. The developed system was implemented in experiments including students from a programming course who were divided into two groups, namely the experimental group and the control group. A system for both groups of students to upload and submit assignments and a code analysis and feedback improvement system were used. Students in the control group only used the assignment uploading and submitting system for basic assignment uploading, verification, and the comparison of test data. After the program was entered, declarative sentence disassembly and dynamic slicing were suggested. Data were sent to GNU Debugger (GDB) and Valgrind for spectral error location; the classification and recording of error types; and the interpretation of the number of error lines, error types, and related variables. Feedback and a generated report were sent back to the student interface to provide effective and useful feedback to the students in the experimental group for them to revise their homework and record the types and number of errors they made in that week's homework in the database. The answers provided by the students to the questions were recorded. The analysis of the pass rates of the students in the experimental and control groups for each homework test aided the understanding of the differences in the learning success of the two groups of students each week. The weekly pass rates and the numbers of measured errors in the experimental group compared with in the control group were input into a distribution map to allow us to better understand whether there was any positive correlation between the detected information, feedback to the students, pass rates of the tests, and other related data. The system statistically obtained feedback and the degree of improvement of homework programs; then, it distributed specially designed questionnaires to all students to directly obtain and quantify their feedback and perceived benefits of this system, thereby verifying the effectiveness of the system and its practicality.

**Keywords:** program analysis; automated debugging; fault localization; dynamic slicing; debug; questionnaire; statistical analysis

## 1. Introduction

The ability to write programs has recently gained importance. Nearly 30 countries have formulated policies in this regard, hoping to strengthen the information capabilities of their citizens from an early age to improve the overall competitiveness of the country. In

2017, Japan proposed the "Future Investment Strategy 2017" and planned to incorporate programming education into the curricula of the compulsory education stages of primary and secondary schools from 2020 as well as further improve the digital teaching materials and evaluation system [1]. The UK also incorporated programming into the curriculum in 2014. There, children start to learn the basics of programming at the age of five, and by the age of eleven they must have the ability to use two programming languages. Besides, universities require at least half of the undeclared students to study programming within five years before graduation to prepare to learn about and apply their knowledge to artificial intelligence.

In a society heavily reliant on information and electronic devices, letting the public understand the principles of program operation could help people to avoid being helpless with respect to information technology (IT). However, not everyone knows how to write programs, but with the aid of basic computational concepts, it is possible to think about more diverse ways of using IT and the Internet. Students can also obtain better logical thinking from correct programming education, thus enhancing their creativity and preparing for solving many related problems in their future.

Learning programming is crucial, but only a few people know how to write programs in the IT departments of universities; only rarely, one can write a system that can be used. One of the problems is that students have too little practical experience, and writing good, stable, and reliable programs requires time-consuming trial-and-error processes and failure-related frustration. A good programming support platform is important to help students overcome the obstacles of programming. Patil et al. [2], Paiva et al. [3], and Jung et al. [4] reported that to reduce the development time or realize the submission of an assignment in a given period, students/programmers widely use social media, especially Google, to find recommendation systems that can suggest a program according to their requirements. The efficient, accurate, and fast development of a code or an assignment depends upon the accurate recommendation given by the system. Wrong recommendations lead to inaccurate software/assignment development, as well as a wastage of time for the student/programmer. Other approaches involve an assessment conducted by students themselves against reference implementations, each with their own mutations, to assess how many of the mutations students can identify in the tests; this naturally encourages students to check their solutions more accurately, rather than have them develop an over-reliance on auto graders [5]. Patil et al. [2] and Lucas et al. [6] proposed both multiple-choice feedback in an online quiz system and the automated assessment of student programming tasks. Using these systems, students can submit their programming assignments numerous times before a deadline and obtain feedback for further improving their code or for fixing mistakes, promoting just-in-time learning (Joan et al. [7]; Yana et al. [8]; Sychev et al. [9]). These systems also allow instructors to save substantial time in terms of manually grading programming assignments and to focus on the pedagogical design of courses.

Although previous research has contributed to the development of automatic feedback, few studies have empirically explored designing formative feedback for programming assignments. If there is only right or wrong evaluation feedback, students are under pressure due to insufficient skills, improper time arrangement, low interest, and other factors, and these pressures are important reasons why students find it difficult to learn programming well. This research study analyzes the errors that students are prone to make when writing programs in programming courses and uses the GDB and Valgrind tools to implement dynamic analysis techniques for their eventual application to student-made programs. Ways of providing good feedback and giving corresponding exercises can significantly improve the success rate of students by giving them the chance to correct their own programming mistakes as compared with only providing correct or wrong feedback.

In this study, we propose to develop a code debugging and feedback system to achieve three objectives. (1) The system is designed to provide teachers with quick programming assignments according to the teaching progress, automatically correct grading, and allow teachers to provide more students with practical programming experience in one course,

thus reducing the pressures of producing and grading a large number of programming tests on teachers. (2) It is designed to be used by students to upload programming assignments and quickly report errors in their programs. According to the logic, grammar, structure, and algorithm used by students, as well as their programming abilities, various types of useful feedback and revision directions can be automatically prompted, which can greatly strengthen the programming design abilities and learning motivation of students, as well as the overall learning effect. In this way, the goal of half of the college students learning programming can be achieved. (3) Lastly, the system is designed to generate relatively different types of programming questions according to the progress of the class, which can provide teachers with quick and accurate programming exercises and can carry out progressive learning according to the difficulty of the exercises.

In this paper, Section 2 presents the concepts and technologies used in this study, Sections 3 and 4 present an analysis-based approach for fault detection and feedback, and Section 5 presents the architecture of the system, experimental process, and results. Then, the conclusions of this study are presented in Section 6.

## 2. Research Background

### 2.1. Control Flow Graph

The control flow graph [10,11] is a directed graph mainly used for static analysis and can display all branches during the execution of programs; it contains if-then-else, while, and do-while controls (Figure 1). It can easily encapsulate the information per each basic block and can locate the inaccessible codes of a program.

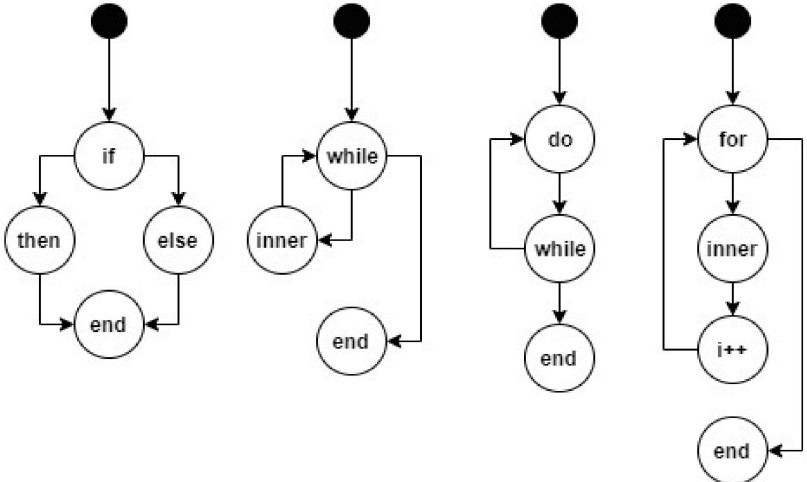

**Figure 1.** Flowchart. From left to right: if-then-else and do-while controls.

### 2.2. Dynamic Dependence Graph

The dynamic dependence graph converts the results of dynamic slicing into graphics [4]. Its advantage is that it makes it easy to find the for-or-while statement in dynamic slicing and to iterate several times, and it clearly indicates the statements affected by each iteration. A primary limitation of this graph is that if there are too many sentences in dynamic slicing, it is complicated. A reduced dynamic dependence graph can be used to simplify the dynamic dependence graph [12,13].

### 2.3. Spectrum-Based Fault Localization

Spectrum-based fault localization uses test cases and their corresponding code coverage to evaluate each program element to determine the possibility of errors in the program. Spectrum-based fault localization was first applied in botanical studies by Jaccard [14] in 1901. Since then, many researchers have used it in other fields (including software debugging) and further improved it. In 2017, Tang et al. [15] compiled the currently used spectrum-based fault localization technology, and in 2018, Javier et al. [16] selected

4 methods with the best results from the existing 18 spectrum-based fault localization methods. The ranking order of these methods was selected to be Kulczynski et al.'s [17], Wong et al.'s [18], Ochiai et al.'s [19], and Janssen et al.'s [20] methods. Fault localization techniques on real vs. artificial faults were compared and reported for the first time by Pearson et al. [21]. It was found that techniques that localized artificial faults best did not perform well on real faults. Xie et al. [22] proposed spectrum-based fault localization to represent the effectiveness of the risk evaluation formula framework based on the concept that the determinant of the effectiveness of a formula is the number of statements with risk values higher than the risk value of the faulty statement. Xie et al. [23] demonstrated that SBSE could be used to automatically design such a formula by recasting this as a problem for genetic programming.

The system detects the wrong program and the program spectrum; records the path of the program in the test; blocks the statement of the program, which is relatively active when the test fails; and applies the formula to calculate each statement or program area in the program. In the error probability score for the block, the higher the score is, the higher the probability of an error is when the program is executed. Its formula usually contains:

1. $N_f(e)$. The test successfully executes line e of the program, and the test result is the number of failures;
2. $N_f(\bar{e})$. The test does not execute line e of the program, and the test result is the number of failures;
3. $N_s(e)$. The test successfully executes the e-th line of the program, and the test result is the number of successes;
4. $N_s(\bar{e})$. The test does not execute line e of the program, and the test result is the number of successes.

Using these factors, the formula is as follows:

$$\text{Jaccard's formula} = \frac{\text{Nf(e)}}{\text{Nf(e)} + \text{Nf}(\bar{e}) + \text{Ns(e)}}.$$

### 2.4. Program Slicing and Decomposition

Program slicing comprises two techniques, static slicing and dynamic slicing [24–26]. The static slicing proposed by Mark Weiser refers to selecting a specific variable V and separating and testing the program statements that make it functional on the premise that it does not affect the overall program behavior. It is often used in maintenance and verification testing.

Taking Figure 2 as an example, the results of the static slicing of X, Z, and TOTAL are shown in Figure 3.

```
1    BEGIN
2    READ(X, Y)
3    TOTAL := 0.0
4    SUM := 0.0
5    IF X<=I
6    THEN SUM := Y
7    ELSE BEGIN
8    READ(Z)
9    TOTAL := X*Y
10   END
11   WRITE(TOTAL, SUM)
12   END.
```

**Figure 2.** Static slicing sample code.

| Performing static slicing of X | Performing static slicing of Z | Performing static slicing of TOTAL |
|---|---|---|
| BEGIN<br>READ(X, Y)<br>END | BEGIN<br>READ(X, Y)<br>IF X < 1<br>   THEN<br>ELSE<br>   READ(Z)<br>END | BEGIN<br>READ(X, Y)<br>TOTAL := 0<br>IF X < 1<br>   THEN<br>ELSE<br>   TOTAL := X<br>* Y<br>END |

**Figure 3.** Result of static slicing of X, Z, and TOTAL.

Agrawal et al. presented the idea of dynamic slicing [27], which involves selecting and slicing sentences that affect or change the content of variable V in the program, with the aim of achieving slicing analysis more efficiently than with static slicing.

In Figure 4, the dynamic slicing path when variable N = 2 is {1, 2, 3, 4, 51, 61, 71, 81, 91, 62, 72, 82, 92, 63, 10}. Program decomposition [28] occurs because of the limited effectiveness of program analysis, and the algorithms used by developers usually have fixed signs to follow. Therefore, this algorithm is designed to statically decompose the program into multiple tokens. The program is orderly decomposed into various components, by which the designed module program disassembles, analyzes, and checks the relationships between each semantic fragment.

```
1  BEGIN
2      READ(N);
3      Z := 0;
4      Y := 0;
5      I := 1;
6      WHILE (I <= N) DO
7          Z := f1(Z, Y);
8          Y := f2(Y);
9          I :=I+1; END WHILE;
10     WRITE(Z);
11 END.
```

**Figure 4.** Dynamic slicing sample code.

*2.5. Software Testing Technique*

A software testing technique is used to test cases or the software interface in various aspects and the correctness of the execution surface of a specific function of the software. It is generally divided into three techniques, white box, gray box, and black box [29].

White box testing performs corresponding tests on the structure of the code and its logic. Since the internal code is in a visible state, it can improve the test efficiency and coverage and ensure the quality of the algorithm by deleting unnecessary code blocks. The disadvantage is that it consumes more human resources and time.

Black box testing performs functional tests corresponding to the requirements. It only needs to execute software, input or perform specific actions according to the requirements,

and confirm that the output or feedback meets the user's expectations. The advantage is that it requires fewer human resources and less time, while the disadvantage is that it cannot be implemented or tested in detail, unlike white box testing.

Gray box testing is a combination of the above two techniques. Certain parts of the code are visible, and the tester conducts black box testing on the interface provided by the developer in a way similar to that of the user.

### 2.6. Questionnaire Survey

Questionnaire surveys are mainly used to collect statistical information on a single/single-field-related issue. They are used to collect and statistically analyze information by targeting specific groups of applicants or by large-scale random sampling without targeting specific groups. Online questionnaires or physical questionnaires can be used for surveys. A questionnaire survey is an efficient way to obtain useful information. It is broadly categorized into four types: open-ended questions, closed-ended questions, sequencing questions, and matching questions [30].

### 3. Code Debugging Method Design

### 3.1. System Architecture Diagram

The proposed system is composed of three major subsystems (Figure 5). The dynamic analysis subsystem can dynamically slice and localize the fault in the code. Based on the theory of the program dependency graph, the dependency relationship of the program code is analyzed, and the dependency relationship information is processed by the fault classification subsystem. The fault classification subsystem performs fault classification on the dependency information, compares it according to the existing implemented error category rules, and processes the comparison result to generate a feedback file. The questions assigned by the subject subsystem mark the sorted questions, such as arrays and string processing. The system performs statistical analyses on the questions raised by the students and merges them with the main question.

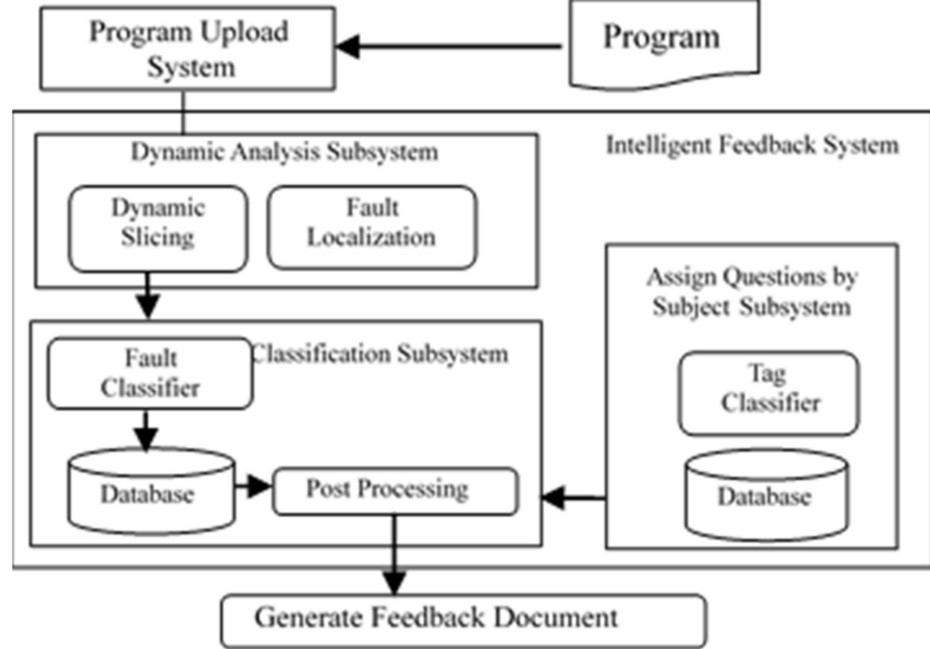

**Figure 5.** Monitoring model architecture.

### 3.2. Pre-Processing of Code Debugging

The debugging method in this study uses dynamic analysis and error location methods, and the fault localization method is mainly based on code coverage. The system inputs a series of test data according to the test case and covers different sentences. The system needs

to filter two kinds of code upload statuses. In a first instance, the first uploaded program fails to compile; because the failed program cannot be executed, it cannot be debugged using dynamic analysis and fault localization methods. Thus, the second program must pass the sample test. In the program uploading system, for each different question type, two different sample test inputs and outputs are provided for the uploader to refer to. The program needs to pass the example test before it can be debugged.

### 3.3. Program Dynamic Analysis

This research study uses the GNU Debugger and Valgrind tools to conduct further analyses and generate feedback on the C language used by students in the classroom, as well as dynamically analyze student-made programs for the types of errors that students are prone to make. GDB can track the execution process of programs and can print out the value and address of any variable. Valgrind mainly debugs the state of memory configuration in the program. In addition, we establish rules for five common mistakes and further explore whether the automatic feedback system can help students pay more attention to the parts that can be easily overlooked before writing the program.

The following are the types of mistakes students tend to make:

1.  An action that attempts to compare or assign a value to an uninitialized variable;
2.  The wrong syntax is used for a specific function in the library;
3.  The program outputs a segmentation fault error;
4.  The array index value exceeds its bounds;
5.  The program execution time exceeds expectations or does not break out of the loop.

Specifically, we address five common student errors in the following ways:

1.  The system marks the uninitialized variable flag as a result of the program analysis and prints the error type, the number of error lines, error content, and the variable whose value has been subjected to an access attempt or has been accessed during actual execution;
2.  Even if students are successful with easier, sample test data, when dealing with more rigorous, real test data, errors in the details of a student's program can be detected. The feedback message for students includes a reminder of the number of lines and the correct use of the strcmp function;
3.  During execution, if there is an attempt to access variable c, which is not assigned to the memory address in the sequence, the title prints the error type, line number, and error content of the segmentation fault;
4.  When executing, it might happen that students do not consider the stop condition of the loop on the periphery of the array, so there is no way to meet the input of the operation test data. As a consequence, the index value of the array trying to access the loop gradually increases and cannot be stopped, resulting in an error exceeding the array boundary;
5.  When dealing with more complex test data, students might enter a loop with imperfect conditional judgment, which leads to program execution taking too long, which may take up system performance. It is easy to see why student-made programs might cause this.

These rules can be analyzed and expanded according to different courses, different students' assignments, and different levels of students. Therefore, this paper proposes a rule-based system that can employ preliminary intelligence.

### 3.4. The Method of Analyzing Mistakes in the Program Process

This research study uses sockets to link the debugging system with the program uploading system. The following are the solution implementation methods for the five mistake types: (1) Use of an uninitialized variable: The system dynamically analyzes the program and records the contents of all variables. When a variable is used and not initialized, it is marked as a mistake and moved forward according to statement dynamic

slicing to find the use path of the variable. (2) Array index out of bounds: The system dynamically analyzes the program and records the initial space size of all arrays. When the array is used, if it exceeds the initial space range, it is marked as a mistake. (3) Incorrect use of the library: The system statically analyzes the program. When the user uses the strcmp function, the expressed condition is == 1 or == −1, and it is marked as a mistake. (4) Segmentation fault: The system dynamically analyzes the program and records the memory address request. When an attempt is made to access a memory location that is not allowed to be accessed, it is marked as a mistake. (5) Irrationally long execution time of the program: The program execution time interval is too long when the analysis process exceeds 15 s. The system checks whether the last execution state is an infinite loop, the program is waiting for input data, or the format conversion of input data is wrong. If one of the above three conditions is true, it is marked as a mistake.

## 4. Feedback Message Design and Debugging Method

This study uses dynamic slicing and fault location methods. After the student uploads the program using the homework uploading system, the output is compared with the designed test data, and it confirms whether the program can be sent to the program analysis and feedback improvement system. If it does not meet the standard condition, the results of the student's program are instantly displayed on the student's user interface. If the analysis standard is met, the student's program is sent into this system for dynamic slicing. The system records all the statements in the program, disassembles all the variables, and analyzes the variables one by one. The data dependency among variables is recorded, integrated into variable data access objects, and stored in series by a dynamic array; then, the array is transferred to the fault localization module.

The GDB in the module first performs a path analysis on the contents of the array and tries to compare whether there are faults that meet the filtered condition. If faults are detected, the fault information is stored in the fault information data access object and sent to the fault categorization module. If no faults are found, the dynamic array is sent to Valgrind for analysis to see if any abnormal use of the memory or abnormal conditions occur during the execution of the program. If so, the fault information is stored in the fault information data access object and sent to the fault categorization module. If the fault is still not found, the analysis is ended. When the fault information is passed to the fault identification and categorization module, the fault information and known fault types are distinguished and classified. After categorization is completed, the system retrieves the specific variable in the variable data access object corresponding to the fault, and the fault flag is sent to the result analysis and report production module. The module then stores the fault variable and its cause in the database, stores the fault category in the database together with the student ID, and classifies the completed fault. The feedback report message is simplified for a better reading experience, and the numbers of program lines, fault variables, and easy-to-read fault feedback reports are returned to the student user interface.

The faults that easily occur in programming learning are here categorized into five types: the program execution time exceeds expectations or does not break out of the loop; the wrong syntax is used for a specific function in the library; the array index value exceeds its boundary; the program generates a segmentation fault; an attempt is made to compare or assign a value to an uninitialized variable. The detected faults are classified, and a fault report is generated and sent back to the user interface of the student, as shown in Figure 6.

The system shows information about the problem; the error problem statement is visualized by the student, and the interface requires the student to pass all unit tests before they can submit their answer.

| Test ID | Fault Category | Faulty Line | Fault Detail |
|---|---|---|---|
| 36 | Array Index Out of Bound | 9 | Execution for the first time<br>Array named "array" exceeds its bound<br>It's a one dimentional array<br>It has maximum index value as 4 and<br>current index value is 10 |
| 36 | Array Index Out of Bound | 10 | Execution for the first time<br>Array named "array" exceeds its bound<br>It's a one dimentional array<br>It has maximum index value as 4 and<br>current index value is 10 |
| 37 | Array Index Out of Bound | 9 | Execution for the first time<br>Array named "array" exceeds its bound<br>It's a one dimentional array<br>It has maximum index value as 4 and<br>current index value is 10 |
| 37 | Array Index Out of Bound | 10 | Execution for the first time<br>Array named "array" exceeds its bound<br>It's a one dimentional array<br>It has maximum index value as 4 and<br>current index value is 10 |
| 38 | Array Index Out of Bound | 9 | Execution for the first time<br>Array named "array" exceeds its bound<br>It's a one dimentional array<br>It has maximum index value as 4 and<br>current index value is 10 |
| 38 | Array Index Out of Bound | 10 | Execution for the first time<br>Array named "array" exceeds its bound<br>It's a one dimentional array<br>It has maximum index value as 4 and<br>current index value is 10 |

**Figure 6.** Feedback of array index out of bound.

## 5. The Solution Implementation Method

### 5.1. System Description

In programming courses, students often do not understand why their programs do not pass a test. Although each student has a different programming style, the mistakes that students make in programming are fundamentally the same, such as the array exceeding the index or the use of uninitialized variables. Therefore, in this research study, we use two mistake locating methods to classify mistakes in students' programs and mark the corresponding program problems. Students can be reminded in advance when making the same mistakes in the future.

The proposed system analyzes the C programming language. Therefore, to avoid adverse student attacks on the system, this system uses gVisor, open-source software by Google, as the system's Sandbox. gVisor implements more than 200 Linux System Calls in the User Space to improve the system's performance safety.

### 5.2. System Process

Figure 7 shows the system flow chart of the program debugging and feedback system. After the students upload their program, the system compares the program with the test data and sends the result to the debug filter. The system uses two debugging methods. Then, the system stores the program's debugging results in the database, classifies the mistake results, marks the corresponding topics according to the classification, and sends the results to the front-end system.

The error localization program in this study mainly performs spectral error localization and dynamic slicing to identify the first five common errors explained as follows:

1. Dynamic analysis subsystem: The dynamic analysis subsystem corresponds to the error location module. The error location module packages the variable relationship in each line of statements into Variable DAO and concatenates objects with ArrayList to form a set of dynamic dependency path structures:

    1-1 Gcov. In the error location module, the Gcov tool is used to analyze the code to obtain the code coverage, and the spectral error location is used to calculate

the similarity coefficient according to the code coverage; finally, the similarity coefficient is calculated. Data are stored in the array;

1-2 Valgrind. In the error location module, the Valgrind tool is both used to analyze the code to obtain memory leaks and reports of illegal uses of memory and to analyze the report content to locate the location of illegally used memory;

1-3 GDB. In the error location module, the GDB tool is used to analyze the code, track the execution path of the program step by step, store the dependencies of each line of statements, and store them in ArrayList;

1-4 Error location module. The error location module packages the similarity coefficient, Valgrind report, and dependencies into ArrayList<VariableDAO> and sends the object to the error identification module.

2. Error classification subsystem: The error classification subsystem corresponds to the error identification module, the error classification and labeling module, and the analysis result module. The error identification module identifies whether there are common errors in the dynamic dependency path structure and retrieves the wrong ones. The object is sent to the error classification and tag module, which error-classifies and tags the question and code, converts the classified Variable DAO to ErrorReportDAO, and sends the object to the analysis result module. Corresponding feedback is given according to the type of classified data:

2-1 Error identification module. The error identification module identifies errors according to the data transmitted by the error positioning module, compares the dynamic analysis structure to see if there are any of the five potential types of error rules, and sends the comparison results to the error classification and tag module;

2-2 Error classification and marking module. The error classification and marking module is classified according to the comparison results; different error rules are classified into different error categories, and the Variable DAO identified as the error is converted into ErrorReportDAO; finally, the object is sent to the analysis result module;

2-3 Analysis result module. The analysis result module translates the error report into format and content that are easy for users to understand and, finally, sends the result to the uploading system through the socket.

3. Thematic question-setting subsystem: The thematic question-setting subsystem corresponds to the marking function in the error classification and marking module. It marks different title-type labels for different program categories, and the error classification subsystem generates questions according to the theme. The label of the subsystem and different statistical reports are calculated:

3-1 Error classification and marking module. The topic-based question-setting subsystem is mainly implemented for the marking function in the error classification and marking module, which marks different programs according to the type of questions marked by the teacher, so that the homework uploading system can perform statistics for different question types and different error categories.

*5.3. Program Experiment and Results*

This research study took programs uploaded by the students of a programming course over the past two years as examples of debugging, modified the feedback for debugging, and re-uploaded the modified programs.

1. Debugging examples involving the use of an uninitialized variable

The use of an uninitialized variable means that the student used a variable without initializing it when declaring the variable. The experimental process is reported below. The subject of the experiment is shown in Figure 8 as an example.

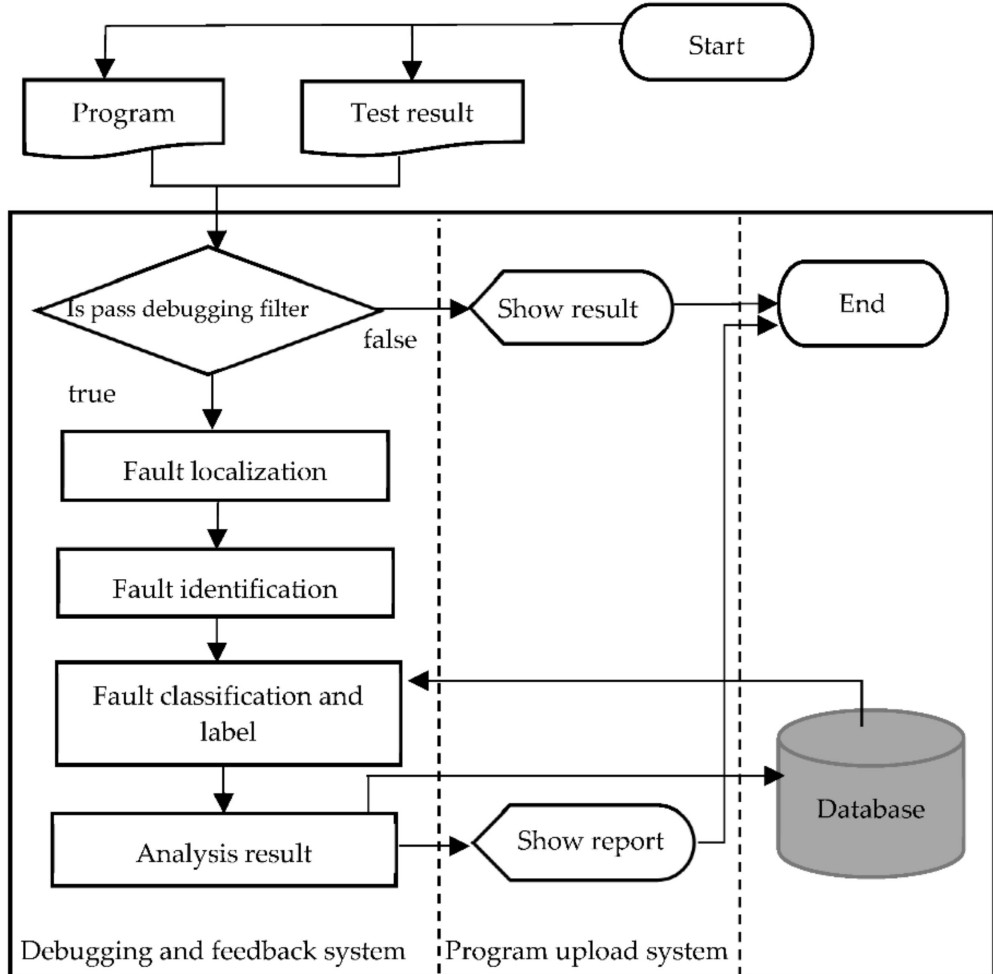

**Figure 7.** System flow chart.

Calculate the addition, subtraction, multiplication and division of two fractions.

Enter four lines, the first and second lines each represent a fraction, the third line represents mathematical operation symbols, and the fourth line represents whether to continue the operation

**Figure 8.** Test question (1).

After uploading the sample program in response to this question in the experiment, the test result is presented as shown in Figure 9.

From feedback report (1) in Figure 10, in test No. 129, an uninitialized variable sign was used code in lines 35 and 51 of the code, and error content was observed in line 19 of the code. The variable sign had been declared, but the definition of code line 21 of the code had not been executed, so the execution of code lines 35 and 51 of the code used the definition of code line 19. Figure 11 below is an example of such a program (program (1)).

| Test Number | Test Results |
|---|---|
| 123 | Test Passed |
| 124 | Test Passed |
| 125 | Test Passed |
| 126 | Test Passed |
| 127 | Test Passed |
| 128 | Test Passed |
| 129 | Test Failed |
| The pass rate : 85% | View Test Report>>> |

**Figure 9.** Test result (1).

According to Figure 4, test No. 129 was not passed. The feedback report is as shown in Figure 10.

| Test Number | Error Category | Error List Line Numbers | Error Message |
|---|---|---|---|
| | | | When executed for the 1st time<br>You used the value of the variable "sing" In the current statement<br>The value is not initialized, its value is: 1<br>The following is the definition and use of this variable |
| 129 | An action that attempts to compare or assign a value to an uninitialized variable | 51 | [def] line 19 int s, n, ik, sign; The variable sign had been declared.<br>[def] line 21 sign=1; The code had not been executed.<br>[use] line 35 if(sign==1){ The variable sign on line 19 is used.<br>[use] line 43 if(sign==1){ The line is not executed.<br>[use] line 51 if(sign==1){ The variable sign on line 19 is used. |
| | | | When executed for the 1st time<br>You used the value of the variable "sing" In the current statement<br>The value is not initialized, its value is: 1<br>The following is the definition and use of this variable |
| 129 | An action that attempts to compare or assign a value to an uninitialized variable | 35 | [def] line 19 int s, n, ik, sign; The variable sign had been declared.<br>[def] line 21 sign=1; The code had not been executed.<br>[use] line 35 if(sign==1){ The variable sign on line 19 is used. |

**Figure 10.** Feedback report (1).

In the example program in Figure 11, as the program did not meet the condition of code line 20, line 21 was definitely not executed, and the content of code line 19 was used in code in lines 35, 42, and 51. In this experiment, after initializing the variable sign to 0 and re-uploading it to the system, the program successfully passed all tests, as shown in Figure 12.

2. Array index out of bounds

When an array of fixed size is declared, an unknown space is accessed if subsequent use exceeds the size of the array. The subject of the experiment is shown in Figure 13 as an example.

The test result after uploading the sample program in response to this question is shown in Figure 14.

The feedback report in Figure 15 shows that all four test failures had the same cause. Figure 15 only shows the feedback content of test No. 11.

```
16    void reduce(int fz,int fm)
17    {
18
19        int s,n,ik,sign;
20        if(fz<0||fm<0){
21            sign=1;
22            fz=abs(fz);
23            fm=abs(fm);
24        }
25        if(fz==0){
26            printf("0\n");
27        }
28        else{
29            s=gcd(fz,fm);
30            fz/=s;
31            fm/=s;
32            n=fz/fm;
33            ik=fz%fm;
34            if(n==0){
35                if(sign==1){
36                    printf("-%d/%d\n",ik,fm);
37                }
38                else{
39                    printf("%d/%d\n",ik,fm);
40                }
41            }
42            else if(ik>=1){
43                if(sign==1){
44                    printf("-%d(%d/%d)\n",n,ik,fm);
45                }
46                else{
47                    printf("%d(%d/%d)\n",n,ik,fm);
48                }
49            }
50            else{
51                if(sign==1){
52                    printf("-%d\n",n);
53                }
54                else{
55                    printf("%d\n",n);
56                }
57            }
58        }
59    }
```

**Figure 11.** Example program (1).

| Test Number | Test Results |
|---|---|
| 123 | Test Passed |
| 124 | Test Passed |
| 125 | Test Passed |
| 126 | Test Passed |
| 127 | Test Passed |
| 128 | Test Passed |
| 129 | Test Passed |
| The pass rate : 100% | View Test Report>>> |

**Figure 12.** Test result after modifying program (1).

Long type Integer calculation.

Calculate the addition, subtraction, and multiplication of two numbers with the array implementation of long integers.

**Figure 13.** Test question (2).

| Test Number | Test Results |
|---|---|
| 11 | Test Failed |
| 12 | Test Failed |
| 13 | Test Failed |
| 14 | Test Failed |
| The pass rate : 0% | View Test Report>>> |

**Figure 14.** Test result (2).

| Test Number | Error Category | Error List Line Numbers | Error Message |
|---|---|---|---|
| 11 | Value exceeds the boundaries | 35 | When executed for the 1st time<br>Array 'c' is outside the bounds of the array<br>1st dimension array<br>The maximum index value is 60, the current index value is 61 |
| 11 | Value exceeds the boundaries | 54 | When executed for the 1st time<br>Array 'a' is outside the bounds of the array<br>1st dimension array<br>The maximum index value is 60, the current index value is 61 |
| 11 | Value exceeds the boundaries | 56 | When executed for the 1st time<br>Array 'b' is outside the bounds of the array<br>1st dimension array<br>The maximum index value is 60, the current index value is 61 |
| 11 | Value exceeds the boundaries | 75 | When executed for the 1st time<br>Array 'c' is outside the bounds of the array<br>1st dimension array<br>The maximum index value is 60, the current index value is 61 |
| 11 | Value exceeds the boundaries | 93 | When executed for the 1st time<br>Array 'c' is outside the bounds of the array<br>1st dimension array<br>The maximum index value is 60, the current index value is 122 |

**Figure 15.** Feedback report (2).

The feedback report shows that in test No. 11 the array with index values of 0~60 code in lines 35, 54, 56, and 75 of the code exceeded the boundary, while the code in line 93 exceeded the 0~121 boundary. Figures 16 and 17 below show the example code.

In label (c) of Figure 16, the array was declared with MAX in code lines 28 and 51 of the code, where MAX is the macro in the third line of the code in label (a). Its value was 61, but the index value started from 61 when the array was used in the code in lines 35 and 75. In this experiment, the conditional expressions in lines 34 and 74 of the code were modified to i < MAX − 1. Similarly, line 82 of the code in label (a) of Figure 17 declared a 61*2 array, but the line started with the index value of 122 in line 93. The initialization of line 92 of the code was changed to i = 2*MAX − 1.

In label (c) of Figure 16, arrays a and b in lines 54 and 56 of the code are parameters. Line 100 of the code in Figure 17b declared arrays a and b, and the code in lines 106, 111, 117, and 123 passed by having a and b as parameters. Obviously, the same problem as above was also found in lines 54 and 56. The initializations of lines 53 and 55 of the code in label (c) of Figure 16 were changed to lenA = MAX − 1 and lenB = MAX − 1. All error lines in the error feedback report in Figure 15 were identified and uploaded again as shown in Figure 18.

```
1    #include <stdio.h>
2    #include <string.h>
3    #define MAX 61
```

(a)

```
26   void add(char a[], char b[]){
27       int i = 0, carry = 0;
28       int c[MAX] = {0};
29       for (i = 0; i < MAX; i++){
30           c[i] = a[i] + b[i] + carry;
31           carry = c[i] / 10;
32           c[i] %= 10;
33       }
34       for (i = MAX; i >= 0; i--)
35           if (c[i] != 0) break;
36       print(c, i);
37       return;
38   }
```

(b)

```
49   void minus(char a[], char b[]){
50       int i = 0, borrow = 0, lenA = 0, lenB = 0;
51       int c[MAX] = {0};
52       char temp[MAX] = {0};
53       for (lenA = MAX; lenA >= 0; lenA--)
54           if (a[lenA] != 0) break;
55       for (lenB = MAX; lenB >= 0; lenB--)
56           if (b[lenB] != 0) break;
57
```

(c)

```
74       for (i = MAX; i >= 0; i--)
75           if (c[i] != 0) break;
76       print(c, i);
77       return;
78   }
```

(d)

**Figure 16.** Example program (2), (**a**) Source code macro definition, (**b**) when the current value in the array indexed starts at 61, (**c**) The declaration of an array with MAX as the size, (**d**) When the current value in the array indexed starts at 61.

3.  Incorrect use of the library

In different environments or versions, when using certain libraries, their usage and return values can be different. If students do not understand the usage of the library in detail before using it, it can lead to unexpected results. This experiment took the strcmp() function in string.h as an example. The subject of the experiment is shown as an example in Figure 19.

After uploading the sample program in response to the question in this experiment, the system reported that line 118 of the code was incorrect and prompted the correct way to use strcmp, as shown in Figure 20.

As shown in Figure 21, the strcmp function was used in line 118, and the condition expression = 1. The return result was 1 when executed in the Windows environment, but the return value was the difference between the ASCII codes of two-character of two-character strings when executed in the Ubuntu environment; for details, refer to the previous publication [15]. In this experiment, after uploading the sample program in

response to this question, the system reported that code line 118 incorrectly used strcmp, and prompted the correct way to use strcmp; the result is shown in Figure 22.

```
80   void multiply(char a[], char b[]){
81       int i = 0, j = 0;
82       int c[2*MAX] = {0};
83       for (i = 0; i < MAX; i++){
84           if (a[i]==0) continue;
85           for (j = 0; j < MAX; j++)
86               c[i+j] += a[i] * b[j];
87       }
88       for (i = 0; i < 2*MAX-1; i++){
89           c[i+1] += c[i] / 10;
90           c[i] %= 10;
91       }
92       for (i = 2*MAX; i >= 0; i--)
93           if (c[i] != 0) break;
94       print(c, i);
95       return ;
96   }
```

(a)

```
98    int main(void){
99        int lenA = 0, lenB = 0, i = 0;
100       char a[MAX] = {0}, b[MAX] = {0};
101       input(a); input(b);
102       lenA = strlen(a); lenB = strlen(b);
103       if (a[lenA-1] == -3 && b[lenB-1] == -3){
104           a[lenA-1] = 0; b[lenB-1] = 0;
105           printf("-"); add(a, b);
106           minus(b, a);
107           multiply(a, b);
108       }
109       else if (a[lenA-1] == -3){
110           a[lenA] = 0;
111           minus(b, a);
112           printf("-"); add(a, b);
113           printf("-"); multiply(a, b);
114       }
115       else if (b[lenB-1] == -3){
116           b[lenB] = 0;
117           minus(a, b);
118           add(a, b);
119           printf("-"); multiply(a, b);
120       }
121       else{
122           add(a, b);
123           minus(a, b);
124           multiply(a, b);
125       }
126       return 0;
127   }
```

(b)

**Figure 17.** Example program (3), (**a**) Line 82 variable declares array size is 61*2, (**b**) Line 100 declares the arrays a and b and passes them as parameters on lines 106, 111, 117, 123.

| Test Number | Test Results |
|---|---|
| 11 | Test Passed |
| 12 | Test Passed |
| 13 | Test Passed |
| 14 | Test Passed |
| The pass rate : 100% | View Test Report>>> |

**Figure 18.** Test result after modifying program (2).

English article analysis.

(1) Replace the P string with the Q string in the article and output it.
(2) Insert the Q string before the P string in the article and output it.
(3) Delete the P string in the article and output it.
(4) Analyze the frequency of all words (word) in the {original} article, and sort them in descending order of frequency.

**Figure 19.** Test question (3).

| Test Number | Test Results |
|---|---|
| | You are using 'strcmp' incorrectly on line 118 of the program, please refer to the following tips. |
| 75 | if Return value < 0 then it indicates str1 is less than str2.<br>if Return value > 0 then it indicates str2 is less than str1.<br>if Return value = 0 then it indicates str1 is equal to str2. |
| | You are using 'strcmp' incorrectly on line 118 of the program, please refer to the following tips. |
| 76 | if Return value < 0 then it indicates str1 is less than str2.<br>if Return value > 0 then it indicates str2 is less than str1.<br>if Return value = 0 then it indicates str1 is equal to str2. |
| | You are using 'strcmp' incorrectly on line 118 of the program, please refer to the following tips |
| 77 | if Return value < 0 then it indicates str1 is less than str2.<br>if Return value > 0 then it indicates str2 is less than str1.<br>if Return value = 0 then it indicates str1 is equal to str2. |
| The pass rate : 0% | View Test Report>>> |

**Figure 20.** Test result (3).

```
114    for(i=0;i<checklen-1;i++)
115    {
116        for(j=i+1;j<checklen;j++)
117        {
118            if(strcmp(array[i],array[j])==1)
119            {
120                for(t=0;t<100;t++)
121                {
122                    char u=array[i][t];
123                    array[i][t]=array[j][t];
124                    array[j][t]=u;
125                }
126            }
127        }
128    }
```

**Figure 21.** Example program (4).

4. Segmentation Fault

This fault occurs when the pointer points to the unallocated memory space and reads it or writes on it. The subject of the experiment is shown in Figure 23 as an example.

| Test Number | Test Results |
|---|---|
| 75 | Test Passed |
| 76 | Test Passed |
| 77 | Test Passed |

The pass rate : 100%      View Test Report>>>

**Figure 22.** Test result after modifying program (3).

Check that 10×10 gomoku can form 5 positions that are connected as a line.

1 means there are chess pieces, 0 means there are no chess pieces

Given a specified board, playing chess at a specific coordinate position (that is, the position 0 changes to 1) can make the five pieces connected in a line, and list all possible coordinate positions

**Figure 23.** Test question (4).

After uploading the sample program in response to this question in the experiment, the test result is shown in Figure 24.

| Test Number | Test Results |
|---|---|
| 188 | Test Failed |
| 189 | Test Failed |
| 190 | Test Failed |
| 191 | Test Failed |
| 192 | Test Failed |
| 195 | Test Failed |
| 196 | Test Failed |

The pass rate : 0%      View Test Report>>>

**Figure 24.** Test result (4).

Figure 24 shows that all tests failed. The feedback report is shown in Figure 25.

| Test Number | Error Category | Error List Line Numbers | Error Message |
|---|---|---|---|
| 192 | Segmentation Fault | 111 | when executed for the 1st time<br>The program illegally accessed the value of variable "r"<br>The memory address is not configured |
| 195 | Segmentation Fault | 111 | when executed for the 1st time<br>The program illegally accessed the value of variable "r"<br>The memory address is not configured |
| 196 | Segmentation Fault | 111 | when executed for the 1st time<br>The program illegally accessed the value of variable "r"<br>The memory address is not configured |
| 188 | Segmentation Fault | 111 | when executed for the 1st time<br>The program illegally accessed the value of variable "r"<br>The memory address is not configured |
| 189 | Segmentation Fault | 111 | when executed for the 1st time<br>The program illegally accessed the value of variable "r"<br>The memory address is not configured |
| 190 | Segmentation Fault | 111 | when executed for the 1st time<br>The program illegally accessed the value of variable "r"<br>The memory address is not configured |
| 191 | Segmentation Fault | 111 | when executed for the 1st time<br>The program illegally accessed the value of variable "r"<br>The memory address is not configured |

**Figure 25.** Feedback report (4).

According to the feedback report, there was a segmentation fault in variable r in line 111 of the code. Figure 26 below shows the relevant parts of the code.

As shown in label (b) of Figure 26, the pch index array in line 261 of the code pointed to the ch two-dimensional array, and the getpass function was called to take pch as a parameter in line 270 of the code. In label (c) of Figure 26, a segmentation fault occurred in line 111 of the code, indicating that the line pointed to unallocated space. According to the previous information, index array r pointed to an array of size 10*10. Then, it affected code line 111 by checking the control and data flow of the code. In the conditional formula of line 110, program $0 \geq x - 1$ was wrong. This error caused index r in line 110 of the code to point to space $x < 0$. Line 110 of the code was modified to if ($9 \geq y + 1$ && $0 \leq x - 1$), and the sample program was re-uploaded. The program shown in Figure 27 passed all tests.

5.  Irrationally long execution time of the program

This means that the program does not end within a defined period. Among at least three situations where the program does not end, the first one is that the program has an infinite loop, which means that the program cannot exit during the loop; the second is that the program stays in the state of waiting for an input; the last situation is that the input format is incorrectly converted. The above conditions mean that the user has not yet understood the question requirements in detail. The subject of the experiment is shown in Figure 28 as an example.

The test result after uploading the sample program in response to this question in the experiment is shown in Figure 29.

In the test shown in Figure 30, the program in the loop of line 25 was wrong and prompted the last executed program code, for example, as shown in Figure 31.

Test 96, shown in Figure 29, failed. The feedback report is shown in Figure 30.

```
1    #include <stdio.h>
2    #include <stdlib.h>
3
4 ●⊟int getpass(int* r[],int y,int x){
5        int a1=0, a2=0, a3=0, a4=0, a5=0, a6=0,
6        a7=0, a8=0, a9=0, a10=0, a11=0, a12=0;
7        int b1=0, b2=0, b3=0, b4=0, b5=0, b6=0,
8        b7=0, b8=0, b9=0, b10=0, b11=0, b12=0;
9        int count, i, j;
10  ⊟    if(x-3>=0){
11  ⊟        if(r[x-3][y]){
12                a1 = 1;
13  -        }
14  -    }
```
(a)

```
254  ⊟int main(){
255        char sc[10][20];
256        int ch[10][10];
257        int i, j,flag=0;
258 ●    int *pch[10];
259  ⊟    for (i = 0; i < 10;i++){
260            gets(sc[i]);
261 ●        pch[i] = ch[i];
262  -    }
263  ⊟    for (i = 0; i < 10;i++){
264            for (j = 0; j < 10;j++)
265                ch[i][j] = sc[i][j*2]-'0';
266  -    }
267  ⊟    for (i = 0; i < 10;i++){
268  ⊟        for (j = 0; j < 10;j++){
269  ⊟            if(ch[i][j] == 0){
270 ●⊟            if(getpass(pch,j,i)){
271                    flag = 1;
272                    printf("%d %d\n", i, j);
273  -            }
```
(b)

```
110  ⊟        if(9>=y+1 && 0>=x-1){
111 ●⊟            if(r[x-1][y+1]){
112                b9 = 1;
113  -            }
114  -        }
```
(c)

**Figure 26.** Example program (5), (**a**) define a 10*10 Gobang disk, (**b**) Describe the pch pointer array points to the ch two-dimensional array, the getpass function is called with pch as the parameter, (**c**) describe 'Segmentation Fault' indicates that the line points to an unconfigured space.

| Test Number | Test Results |
|---|---|
| 188 | Test Passed |
| 189 | Test Passed |
| 190 | Test Passed |
| 191 | Test Passed |
| 192 | Test Passed |
| 195 | Test Passed |
| 196 | Test Passed |
| The pass rate : 100% | View Test Report>>> |

**Figure 27.** Test result after modifying program (4).

Poker.

A, 2, 3, 4, 5, 6, 7, 8, 9, 10, J, Q, K
A~10 points are 1~10, J, K, Q are 0.5.
The computer and the player randomly deal out poker cards, and the total points are close to 10.5 to win.
If it exceeds 10.5, the knockout score is 0 and the party may not continue to ask for cards.
In any round of merger, the party that has not asked for a card loses the right to ask for a card.

The computer will only ask for cards in the following situations:
1. The total number of points is smaller than the player
2. The total number of points is less than 8 points (including 8 points)

**Figure 28.** Test question (5).

| Test Number | Test Results |
|---|---|
| 94 | Test Passed |
| 95 | Test Passed |
| 96 | Test Failed |
| 97 | Test Passed |
| 98 | Test Passed |
| 99 | Test Passed |
| 100 | Test Passed |
| The pass rate : 85% | View Test Report>>> |

**Figure 29.** Test result (5).

The comparison between the sample program and the error feedback report shows that the program continuously executed lines from 25 to 39, which contained many conditional expressions. According to the conditional expression and the requirements of the test questions in Figure 28, the bottom-line question requirements in the test questions in Figure 28 were not written in the sample program. In line 32, the following requirements were met: (1) the total number of points was less than the player or (2) the total number of points was less than 8 (including 8 points), but the essential part of the test question was not processed. Our experiment accomplished this requirement as shown in Figure 32.

This experiment defined the is End variable to meet the needs of the problem and initialized the variable to false. In line 33 of the code in Figure 32, an if statement was added to determine whether player B wanted to obtain another card or not; if false, player B could ask for another card; if true, then line 38 of the code was executed, indicating that

no cards would be requested afterward. In this experiment, the modified sample program was re-uploaded. The program passed all tests as shown in Figure 33.

| Test Number | Error Category | Error List Line Numbers | Error Message |
|---|---|---|---|
| 11 | Program execution time is too long | 35 | You may have neglected a certain requirement, please confirm the conditional judgment in line 25 And all the judgments in the loop, such as if, for, while, etc... Are there any errors? And learn more about the topic requirements, the following is the program error snippet <br><br> line 25 while (q=='Y'){ <br> line 26 if (q=='Y'){ <br> line 4 if (x=='1') return 10; <br> line 5 else if ((x>='2')&&(x<='9')) return (x-'0'); <br> line 6 else if (x=='A') return (1); <br> line 7 else return 0.5; <br> line 31 if (q!='N'){ <br> line 10 if (sum<=8) return 1; <br> line 11 return 0; <br> line 4 if (x=='1') return 10; <br> line 5 else if ((x>='2')&&(x<='9')) return (x-'0'); <br> line 6 else if (x=='A') return (1); <br> line 7 else return 0.5; <br> line 39 getchar(); |

**Figure 30.** Feedback report (5).

```
3   double input(char x, char y) {
4       if (x=='1') return 10;
5       else if ((x>='2')&&(x<='9')) return (x-'0');
6       else if (x=='A') return (1);
7       else return 0.5;
8   }
9   int isDeal(int sum) {
10      if (sum<=8) return 1;
11      return 0;
12  }
13  int main() {
14      char x,y,w,z,q;
15      double in=0, scoreA=0, scoreB=0;
16      scanf("%c%c\n", &x, &y);
17      in = input(x, y);
18      scoreA = scoreA + in;
19
20      scanf("%c%c\n", &w, &z);
21      in = input(w, z);
22      scoreB = scoreB + in;
23
24      scanf("%c\n",&q);
25      while(q=='Y'){
26          if (q=='Y') {
27              scanf("%c%c\n", &x, &y);
28              in = input(x, y);
29              scoreA = scoreA + in;
30          }
31          if(q != 'N'){
32              if (isDeal(scoreB)||(scoreB<scoreA)) {
33                  scanf("%c%c\n", &x, &y);
34                  in = input(x, y);
35                  scoreB = scoreB + in;
36              }
37          }
38          scanf("%c",&q);
39          getchar();
40      }
```

**Figure 31.** Example program (6).

```
 3   double input(char x, char y) {
 4       if (x=='1') return 10;
 5       else if ((x>='2')&&(x<='9')) return (x-'0');
 6       else if (x=='A') return (1);
 7       else return 0.5;
 8   }
 9   int isDeal(int sum) {
10       if (sum<=8) return 1;
11       return 0;
12   }
13   int main() {
14       char x,y,w,z,q;
15       bool isEnd = false;
16       double in=0, scoreA=0, scoreB=0;
17       scanf("%c%c\n", &x, &y);
18       in = input(x, y);
19       scoreA = scoreA + in;
20
21       scanf("%c%c\n", &w, &z);
22       in = input(w, z);
23       scoreB = scoreB + in;
24
25       scanf("%c\n",&q);
26       while(q=='Y'){
27           if (q=='Y') {
28               scanf("%c%c\n", &x, &y);
29               in = input(x, y);
30               scoreA = scoreA + in;
31           }
32           if(q != 'N'){
33               if (isDeal(scoreB)||(scoreB<scoreA) && !isEnd) {
34                   scanf("%c%c\n", &x, &y);
35                   in = input(x, y);
36                   scoreB = scoreB + in;
37               }else
38                   isEnd = true;
39           }
40           scanf("%c",&q);
41           getchar();
42       }
```

**Figure 32.** Modified sample program.

| Test Number | Test Results |
|---|---|
| 94 | Test Passed |
| 95 | Test Passed |
| 96 | Test Passed |
| 97 | Test Passed |
| 98 | Test Passed |
| 99 | Test Passed |
| 100 | Test Passed |
| The pass rate : 100% | View Test Report>>> |

**Figure 33.** Test result after modifying program (5).

## 6. Experiment

### 6.1. System Information

The system used in this study used GDB and Valgrind to assist in analyzing the errors generated by the program, and further analyses and feedback were conducted by

students using the C language. The students in the experimental group used the homework uploading system and the improved system (the system in this study) to obtain debugging feedback, while the students in the control group only used the homework uploading system for general assignment uploading. Students in the experimental group could see the feedback given by the improved system in the homework uploading system.

### 6.2. Results and Comparison of Statistical Analyses

In the system interface, we recorded each log of the code being executed, and following the results of each unit test, the participant ran a unit test. The study was conducted by randomly selecting the experimental group and the control group. The results obtained after the students uploaded the assignments were used for final analyses and statistics. The results included passing the test, failing the test, and the minus sign (indicating no upload result), as shown in Figure 34.

**Figure 34.** Results of the students' programming assignments.

The students who did not answer after the deadline for homework submission or were deleted by the system due to plagiarism could not participate in the statistical analysis of this experiment as they formed a sample that did not meet the statistical inclusion conditions.

This study used Excel to perform data-related statistics and analyses. The difficulty of each week's homework consisted of the introduction of a keyword of the C language and its application. Teaching and practice were carried out stepwise. Figure 35 shows a statistical analysis of the number of questions answered by students in a single week. Right and wrong features were used to count the correctness of the answers of the students who answered each question during the week. Among them, weights were used to make the data more inclined to the results of analyses and statistics when a normal number of students finished the assignments:

$$\text{Pass rate(A)} = \frac{\text{pass amounts of each question(R)}}{(\text{pass amounts of each questions(R)} + \text{fail amounts of each question(W)})} \tag{1}$$

where A is the accuracy, R is the right amount, and W is the wrong amount.

$$\text{Pass rate growth}(\%) = \frac{\text{the pass rate of the experimental group}(\%)}{\text{the pass rate of the control group}(\%)} - 100\% \qquad (2)$$

|  | Passed amount | Failed amount | Answered amount | Pass rate |
|---|---|---|---|---|
| Question 1 | 346 | 4 | 50 | 98.857% |
| Question 2 | 327 | 3 | 44 | 99.091% |
| Question 3 | 343 | 0 | 49 | 100.000% |
| Question 4 | 192 | 0 | 48 | 100.000% |
| | | | | |
| Total | 1208 | 7 | 191 | 99.424% |

**Figure 35.** Statistical analyses of the number of questions answered by students in a single week.

By counting the sum of the number of students passing or failing the tests each week, excluding the students who plagiarized their work and those who did not submit the assignments, the correct-answer rate of the test was calculated on a weekly basis as shown in Figure 35.

We evaluated whether students could effectively understand the code correction prompt provided by the error message through the feedback error message generated by the system after uploading the assignment, make corrections or deletions to fix the errors, and then re-upload and submit the assignment. The average total pass rate of each week's assignments was calculated based on the total numbers of tests passed and failed for each question during the week and the weight of each question. Fault detection recorded the analysis and the number of classifiable errors measured in the code of the students in the experimental group; we observed changes in the relationship between the pass rate and the fault rate of each week's assignments (fault detection formed a special dataset in the experimental group only), as shown in Figure 36.

|  | Question 1 | Question 2 | Question 3 | Question 4 |  | Average pass rate of the week | Fault Detected |
|---|---|---|---|---|---|---|---|
| Week 1 | 98.857% | 99.091% | 100.000% | 100.000% |  | 99.424% | 4 |
| Week 2 | 100.000% | 100.000% | 100.000% | 100.000% |  | 100.000% | 4 |
| Week 3 | 100.000% | 100.000% | 100.000% |  |  | 100.000% | 11 |
| Week 4 | 98.339% | 99.573% |  |  |  | 98.879% | 46 |
| Week 5 | 100.000% | 97.479% | 98.571% |  |  | 98.617% | 116 |

**Figure 36.** Correct-answer rate and average pass rate for assignments each week.

This study also recorded the total pass rate, the total number of questions, the weekly pass rate, the average number of students who answered each question, and other related data to facilitate the grasp of various variables in the statistics and reduce the deviation value after estimations, as shown in Figure 37.

| Total pass rate | Total passed amount | Total failed amount | Experimental group week-by-week total pass rate | pass rate growth of the week |
|---|---|---|---|---|
| 99.479% | 4395 | 23 | 99.429% | 0.030% |
| | | | 99.690% | 0.740% |
| Total question amount | Answered amount | Average answered amount per question | 99.800% | 2.150% |
| 16 | 705 | 44.06 | 99.572% | 8.250% |
| | | | 99.414% | 3.280% |
| | Two groups total answered amount | Two groups average answered amount per question | | |
| | 1422 | 88.875 | | |

**Figure 37.** Total pass rate and various data analyzed in the above-mentioned assessments.

As shown in Figure 38, the pass rates (including weights) of each week of the experimental group and the control group were used to compare and prepare a line chart to understand the difference in the pass rates of the two groups when progressing from simple to more complex course contents. The learning effectiveness of the students in the experimental group was found to be stable, and the pass rate was better than that of students in the control group most weeks.

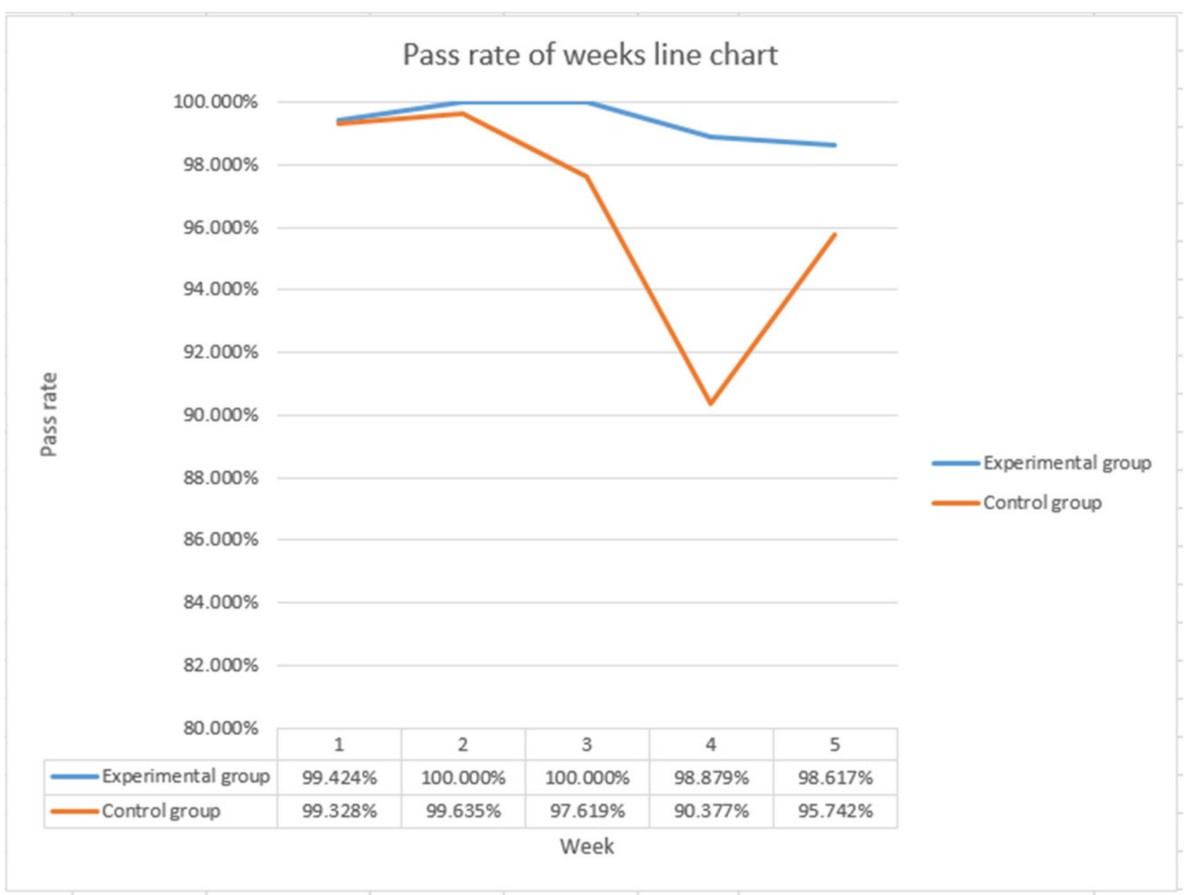

**Figure 38.** Line chart showing the weekly pass rate of the students.

As shown in Figure 39, the total pass rates (including weights) of the experimental group and the control group for each week were used to compare and record the changes in long-term cumulative question pass rates. We did not compare the difficulty of each week's questions but analyzed them statistically based on the cumulative method of each week's questions to understand the impact of the students' weekly answers on the overall pass rate. Figure 39 shows the difference in the learning effectiveness of the two groups of students each week.

As shown in Figure 40, the number of fault feedbacks obtained from the experimental group data and the pass-rate growth were used to calculate the ratio of the feedback given to the experimental group compared to the control group and prepare a scatter chart to obtain a trend showing the relationship between the two; in this chart, the trend shows a positive correlation. According to Figure 40, the number of students in the experimental group who triggered false detections and received feedback during the week was directly proportional to the growth of the pass rate of the experimental group compared to the control group, indicating that the number of false reports detected and the feedback provided to students affected the students' assignment performance, and the impact was found to be positive, thereby indicating an improvement of the learning efficiency. (See Equation (2)).

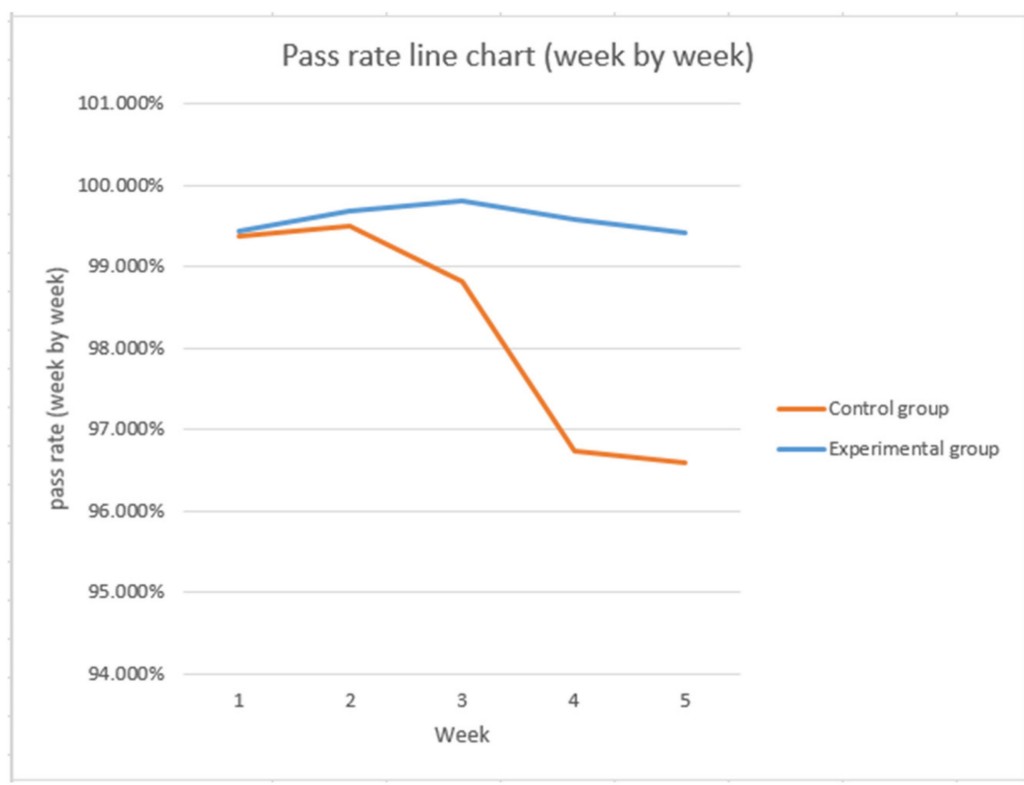

**Figure 39.** Line chart presenting the weekly pass rates of the students.

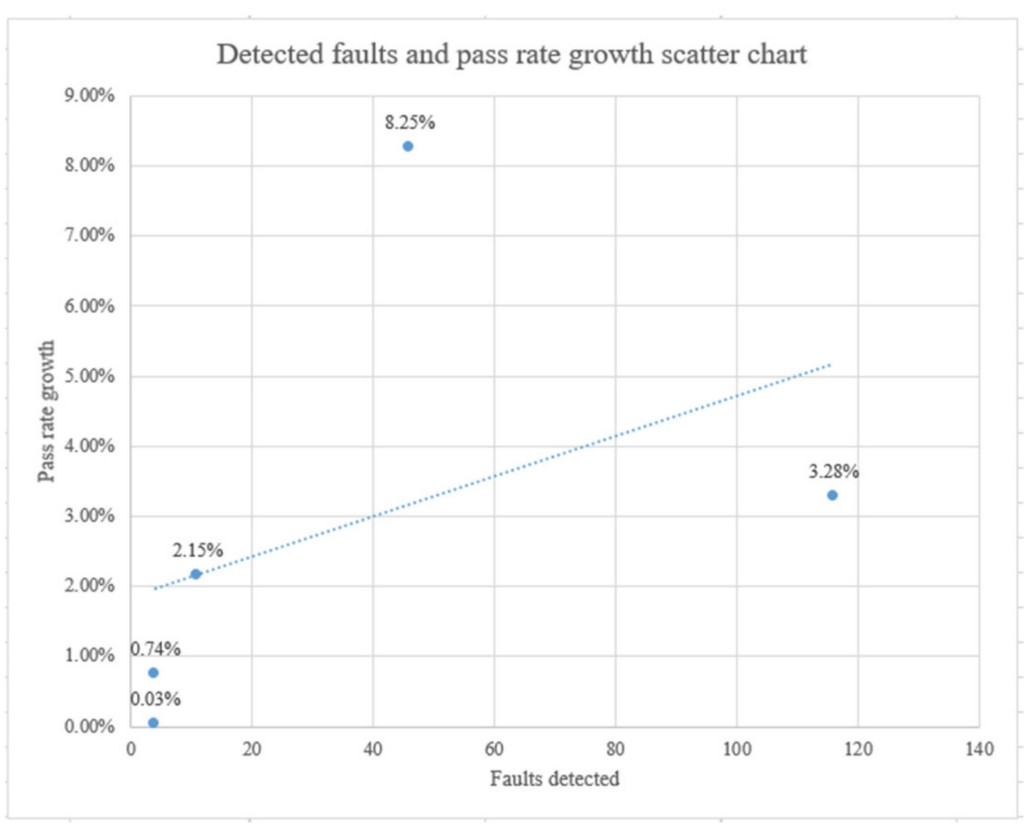

**Figure 40.** Scatter chart to detect faults and pass-rate growth.

### 6.3. Comparison with Other Studies

Table 1 presents a comparison of the experimental results of this study and those obtained by previous, related studies. The five different program feedback systems proposed in this study are better than other previously proposed methods; in addition to the basic test pass and fail, code coverage, and real-time feedback prompt part of program feedback, this research study also adds program analysis and error localization. Before analysis, the system checks each student's program. The declarative sentence is recorded in the file, and the specific test data that cause the error are entered again and tested. The program is first tested with the originally set test data, and the program fails its tests if the system fails to compile the sample test data provided to the students; these two situations are excluded. After obtaining the coverage of the test and code, the error is located, and the results of debugging are used in GDB and Valgrind to perform dynamic analyses. After the error category is detected, it is labeled and transferred to the database for records, and simultaneously, its corresponding text message is sent back to the homework submission system, which is displayed in the student's error inspection report interface.

**Table 1.** Comparison of the findings of this study with those observed in other studies.

| Code Error Feedback Method | This Paper | Lucas et al. [6] | Joan et al. [7] |
|---|---|---|---|
| Test failure vs. test pass | Yes | Yes | Yes |
| Code coverage | Yes | Yes | Yes |
| Feedback analysis interface | Yes | Yes | No |
| Program analysis and program error location | Yes | Yes | No |
| Feedback message after debug feedback function | Yes | No | No |

### 6.4. Questionnaire Survey Statistics

According to Figure 41, half of the students who filled out the questionnaire received non-output fault feedback. Those who did not receive this feedback could only trigger the system feedback when their program did not meet the input conditions, which led to the extension of the execution time. By assessing the satisfaction statistics, the overall satisfaction degree of the students who completed the paper for the improvement of the system's work correction speed was found to be 1.03, which indicates a degree of partial satisfaction.

| | |
|---|---|
| Experimental group total students | 54 |
| Experimental group students who filled the questionnaire | 38 |
| Student amount of Obtaining non-input fault feedback | 19 |
| Student amount of Not obtaining non-input fault feedback | 19 |
| | |
| Student amount of satisfying for improving assignment | |
| Strongly agree(3) | 10 |
| Agree(2) | 9 |
| Slightly agree(1) | 6 |
| Normal(0) | 6 |
| Slightly disagree(-1) | 2 |
| Disagree(-2) | 2 |
| Strongly disagree(-3) | 3 |
| | |
| Questionnaire filled rate | 70.37% |
| Average agreement score of Satisfaction | 1.03 |

**Figure 41.** Student questionnaire survey results.

## 7. Conclusions

This study proposes the use of the GDB and Valgrind tools to implement dynamic slicing by integrating them with spectral error localization technology. The similarity coefficient of the error localization analysis of the program was used to improve the performance of dynamic slicing, and the data flow and control flow were analyzed after slicing to obtain the results. Then, a dynamic dependency graph of the analyzed results was prepared. The system then analyzed whether there were common errors in the program according to the dynamic dependency graph and sorted the feedback information based on the analysis results to provide students with the direction to follow for modifying their program to correct the errors.

Most of the automatic correction systems on the market usually only provide simple information such as test failures, test passes, and compilation errors as feedback for programs uploaded by students. This simple information is often not very helpful for students attempting to successfully debug their programs. The sample questions mentioned in this study showed the correctness and incorrectness of the uploaded programs, located the specific error condition, and provided corresponding feedback. In addition, the study utilized group experiments to give students appropriate feedback; moreover, we used various data, statistical analyses, and a questionnaire specially designed for students to understand the difference in the progress of two groups of students as well as the effectiveness and practicality of the system.

Finally, here we provide evidence that implementing the problem solutions proposed in this study can be effective in helping students who often struggle with coding. We believe that tools that help to reduce this cognitive load can aid teaching assistants in helping students to learn more effectively.

**Author Contributions:** Conceptualization, J.-Y.K., P.-F.W. and H.-C.L.; methodology, J.-Y.K., P.-F.W. and H.-C.L.; software, J.-Y.K. and H.-C.L.; validation, J.-Y.K. and H.-C.L.; writing—original draft preparation, J.-Y.K. and H.-C.L.; writing—review and editing, J.-Y.K., H.-C.L. and Z.-G.N. All authors have read and agreed to the published version of the manuscript.

**Funding:** This research was funded by [National Taipei University of Technology- Beijing Institute of Technology Joint Research Program] grant number [NTUT-BIT-108-01].

**Institutional Review Board Statement:** Not applicable.

**Informed Consent Statement:** Not applicable.

**Data Availability Statement:** Not applicable.

**Conflicts of Interest:** The authors declare no conflict of interest.

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
