# Peer review of "A Feedback System Supporting Students Approaching a High-Level Programming Course"

_applsci, doi:10.3390/app12147064_

Round 1

Reviewer 1 Report

Comments have been taken into account. However, no many references have been added.

Author Response

Point 1:
Comments have been taken into account. However, no manyreferences have been added.

Response 1:
We would like to first thank the editor and reviewers for their response to our work and their constructive comments. We have added the References as you suggested (Joan et al. (2022), Yana et al. (2022), Sychev et al . (2022), and Jung et al. (2022) papers in addition to the more recent papers. (Page 26; red text).

We appreciate for Editors and Reviewers’ warm work earnestly and hope that the correction will meet with approval.

Reviewer 2 Report

The manuscipt revised. The process of testing students has become clear, it is clear what kind of errors and how the system processes them. But

1. I still can not find scientific novelties.

2. Conclusions are week.

3. I still think that the words "intelligent feedback" implies the use of machine learning methods such as neural networks etc. But this is not described in the article. I would just call it "feedback"

Author Response

Point 1:
I still can not find scientific novelties.

Response 1:
We are grateful for your consideration of this manuscript, and we also very much appreciate your suggestions, which have been very helpful in improving the manuscript.
We add in Section 1 that the feedback system proposed in this study is combined with spectral error localization techniques,Errors in dynamic analysis can be found more accurately, and the statements and structures of code uploaded by students can be decomposed and analyzed. (page 2, paragraph 4; red text).
We mentioned in section 7 that most of the general learning platforms and environments only provide program results, usually only simple information such as test failures and test passes, and compilation errors. This simple information is often not very useful for students to debug help. Based on the above questions, this research not only shows the correct and incorrect feedback but also can locate the error for a specific error condition and give the corresponding feedback.
Finally, we presented evidence through group experiments, and give appropriate feedback to the students in the experimental group having difficulty when helping a student with their code may help more effective in helping students.

Point 2:
Conclusions are weak.

Response 2:
We agree with the reviewer's comment and we have now revised the conclusion to reinforce this research, the proposed programming debugging feedback systems can help students understand and solve problems in the C Program curriculum. (page 25, paragraph 3; red text).

Point 3:
I still think that the words "intelligent feedback" implies the useof machine learning methods such as neural networks etc. Butthis is not described in the article. I would just call it "feedback".

Response 3
We would like to thank the editor and reviewers for their response to our work and their constructive comments.
We will supplement in Section 3.3 that this study proposes automatically generated code corrections to compare and give error message feedback by automatically evaluating the code submitted by students when they upload their assignments. We propose a five-point return rule-base to explore whether the feedback system in this study is effective in helping students solve most problems and move on to the next unit. (page 7, paragraph 2; red text).
-4-
In this paper, we supplement in Section 5.2 that the programming debugging and feedback system flow, by error localization program mainly performs spectral error localization and dynamic slicing for the first five common errors. (page 10, paragraph 1; red text).
Below is our brief supplemental description:
We collected five common mistakes students made in the past two years to discuss,
1. An action that attempts to compare or take a value on an uninitialized variable.
2. The wrong syntax is used for a specific function in the library.
3. The program outputs Segmentation fault type error.
4. The array index value exceeds its bounds.
5. The program execution time exceeds expectations or does not break out of the loop.
This study addresses the following common student errors:
1. The system marks the uninitialized variable flag on the result of the analysis program, and prints the error type, the number of error lines, the content of the error, and the value in the variable that was attempted to be accessed or was accessed during the actual execution.
2. Even if it passes the relatively easy sample test data, in the more rigorous real test data, errors in the details of the student's program can be detected, and the feedback information to the student includes the line number reminder and the correct use of the strcmp function.
3. During the execution process, try to access the variable c that is not allocated to the memory address in the sequence, so the error type, line number, and error content of Segmentation Fault will be displayed in the title.
4. When using, do not pay attention to the stop condition of the loop around the array, and cannot meet the input of the operation test data, so that the index value of the array trying to access the loop gradually increases and cannot be stopped, resulting in an error beyond the boundary of the array.
5. In the more complex test data, entering a cycle with imperfect condition judgment leads to long program execution time, which may occupy system performance. It's easier to understand why their program might be causing this.
These rules can be analyzed and expanded according to different courses, different students' assignments, and different levels of students. Therefore, this paper is a rule-based system, which can have preliminary intelligence.

We appreciate for Editors and Reviewers’ warm work earnestly and hope that the correction will meet with approval.

Round 2

Reviewer 2 Report

The article has been revised. My remarks are taken into account

Author Response

Thank you very much for your previous comments that helped us improve this manuscript.

This manuscript is a resubmission of an earlier submission. The following is a list of the peer review reports and author responses from that submission.

Round 1

Reviewer 1 Report

The article presents an approach for checking student programs for correctness. For this, tools such as gdb and valgrind are used.

There are the following comments:

1. Judging by the description, the system is designed to check only the simplest C programs containing loops and conditions. But, for example, there are no such basic checks as the use of pointers, the correct use of elements of object-oriented programming, etc. It is necessary to give an example of a typical task so that it is clear to the reader which tasks are being discussed. But I think that the functionality is poor.

2. The system contains a module that checks the source code for plagiarism. But the simpler the program, the fewer ways you can solve it. And because of this, some programs can be considered plagiarism by mistake.

3. It seems to me that part of the classes should be devoted to studying the gdb and valgrind. It will be more useful for students in the future.

4. There is no comparison of functionality with analogues. There are many similar systems on the market, including free ones.

5. It is not clear what is the scientific novelty of the work.

6. The word "intelligent" in the title of the article needs an explanation. The work does not use AI.

Reviewer 2 Report

Dear Authors,

I appreciate your work.

It has been correctly presented state-of-art, but it is not enough.

The experiment is very unclear.

No standardized example programs analyzed.

GDB gives a very clear indication of the use of an uninitialized variable. I don't see the need to create additional tools for this purpose.

The analysis of a reference to an element outside the array scope is more interesting. Unfortunately, it is about statically declared array. Such an array is not a challenge.

Valgrind handles dynamic arrays quite well. However, this has not been used by the authors.

What do the authors propose to do with the remaining trivial errors, such as:

- missing ; at the end of a line,

- type incompatibility,

- undeclared function,

- incompatibility of a function prototype with its implementation,

- ...

As a reviewer I have been put in a very uncomfortable situation.

In your work I cannot find novelties worthy of giving a positive rating.

Reviewer 3 Report

Abstract: Is "messy". Starts describing what the study analyses and in the middle of the paragraph the aim of the study is pointed out, to end with the statistical methods.

Research background: Earlier studies are mentioned, but they are about 50 years old. That reference is not valid if not justified that nothing has been done in between.

Spectrum-based fault-localization, on the other hand, is based on more recent research. 

In general, very basic explanation about the methodologies, and although utilization is mentioned, it is not explained why those methods are suitable for those applications.

Students mistakes are pointed (5 of them) and related with 2 years experience. However, it is not clear why those 5 are chosen, their weight over the total or why only 5.

Feedback message design is correctly explained, according to the mistakes described above.

In the experimentation, from the total amount of students 1208, only 191 participated. A table is shows with a few of them, but not sure if they are all of them. In any case, there are not enough "failed" data to extract conclusions.

There is no any statistical analysis to determine if the obtained data is relevant or not.

Conclusions are week.

References: Very few in quantity for a research article. In addition, most of them are not recent.